# Optimal Sensor Placement for Vibration-Based Damage Localization Using the Transmittance Function

**DOI:** 10.3390/s24051608

**Published:** 2024-03-01

**Authors:** Ilias Zacharakis, Dimitrios Giagopoulos

**Affiliations:** 1Department of Mechanical Engineering, University of Western Macedonia, Bakola & Sialvera, 50100 Kozani, Greece; 2Department of Mechanical Engineering, Aristotle University of Thessaloniki, 54124 Thessaloniki, Greece

**Keywords:** sensor placement, damage detection, damage localization, vibration-based optimization, transmittance function

## Abstract

A methodology for optimal sensor placement is presented in the current work. This methodology incorporates a damage detection framework with simulated damage scenarios and can efficiently provide the optimal combination of sensor locations for vibration-based damage localization purposes. A classic approach in vibration-based methods is to decide the sensor locations based, either directly or indirectly, on the modal information of the structure. While these methodologies perform very well, they are designed to predict the optimal locations of single sensors. The presented methodology relies on the Transmittance Function. This metric requires only output information from the testing procedure and is calculated between two acceleration signals from the structure. As such, the outcome of the presented method is a list of optimal combinations of sensor locations. This is achieved by incorporating a damage detection framework that has been developed and tested in the past. On top of this framework, a new layer is added that evaluates the sensitivity and effectiveness of all possible sensor location combinations with simulated damage scenarios. The effectiveness of each sensor combination is evaluated by calling the damage detection framework and feeding as inputs only a specific combination of acceleration signals each time. The final output is a list of sensor combinations sorted by their sensitivity.

## 1. Introduction

Structural health monitoring (SHM) systems have been developed and applied, either in the inspection phase or live monitoring of a structure, in many different industries from civil infrastructure to aerospace and automotive. This trend is linked to economic but also health reasons. These systems can not only lower the cost and time of maintenance and inspection of structures, but also prevent human errors during these processes. Furthermore, undetected damage will progress, as it is not treated properly, and can cause serious injuries and fatal accidents. Burgos et al. presented in their review the major aspects of damage identification in SHM [1]. The current work focuses on vibration-based methods that rely on the fact that structural damage will affect the dynamic characteristics of a structure. Different approaches that belong in this category have been applied in the past [2,3,4,5,6], and methods that can be used in real-time scenarios have also been presented [7].

A damage localization methodology is as good as the information that it has available. An Optimal Sensor Placement (OSP) procedure maximizes the chances of identifying or locating future damage in a structure. The location of the sensors within the structure plays a pivotal role in the accuracy and also the applicability of each method. The OSP problem can be treated as an optimization problem with a final task to find the best set of sensor locations and/or the number of required sensors. Some damage localization methodologies can be used interchangeably with OSP procedures as they rely upon optimization algorithms in order to find the damage in a structure. The methodology that is presented in the current work falls into this category.

Over the years, many OSP methods have been proposed and applied in single-part and complex multi-part structures that might include common or advanced materials such as fiber-reinforced composites. Gomes et al. compared different response metrics with an application in a laminated composite plate [8]. Gomes and Pereira used the Firefly optimization algorithm in a fuselage structure by using the FIM (Fisher Information Matrix) criterion which uses modal data from the structure [9]. Lin, Xu, and Law used the NSGA-II multi-objective algorithm in order to find the optimal placement of multiple types of sensors such as accelerometers, displacement transducers, and strain gauges [10]. Ding-Cong, Dang-Trung, and Nguyen-Thoi used the Jaya optimization algorithm and the Modal Assurance Criterion (MAC) in laminate composite structures [11]. Beygzadeh et al. presented an improved genetic optimization algorithm designed for OSP with a focus on space structures [12]. Pereira et al. presented a multi-objective sensor placement optimization procedure using feature selection with a variable number of sensors [13]. Jung, Cho, and Jeong used Genetic Algorithms (GA) and the MAC on flexible structures for the placement of sensors [14], and similarly, An, Youn, and Kim investigated the number and placement of sensors [15].

Many different optimization algorithms have been proposed for OSP or damage detection purposes. The most commonly used fall in the category of meta-heuristic algorithms. Such algorithms include Genetic Algorithms [16,17,18], Ant Colony Optimization (ACO) [19,20,21], Particle Swarm Optimization (PSO) [22,23,24], Covariance Matrix Adaptation Evolution Strategy (CMA-ES) [25], Firefly Algorithm (FA) [26], Sunflower Optimization (SFO) [27], Bat optimization algorithm [28,29,30], Artificial Bee Colony (ABC) [31], the Wolf algorithm [32], and others. Besides the selection of the optimization algorithms, the dynamic response metric of the structure that is taken into account in the objective function plays a crucial role in the final result. One of the most commonly used metrics is the Modal Assurance Criterion (MAC) [11,14], although others have also been applied successfully, such as the Fisher Information Matrix (FIM) [33,34], Kinetic Energy [35], Eigenvalue Vector Product [36], Information Entropy [37,38], Effective Independence [39], and the Average Driving-Point Residue (ADPR) [36]. Alkayem et al. included in their survey some common optimization algorithms and metrics in this category that are used for damage detection in combination with Finite Element Models (FEM) [40].

The present methodology is based on the Damage Detection Framework that has been presented by the authors in the past [22]. This framework uses the Particle Swarm Optimization (PSO) algorithm [41,42,43], which is a population-based meta-heuristic algorithm and has been demonstrated to perform very well for this task. This framework uses a parametric damaged area which is controlled by the optimization parameters, and changes location and material mechanical properties in order to simulate the effect of the damage. The dynamic response metric that is used is the Transmittance Function (TF) [44]. The TF is calculated between two output acceleration signals and does not require any measurement from the experimental excitation. As such a change in the TF curve represents a change in the structural properties, it is a sensitive metric for damage detection [44,45,46]. Since the calculation of the TF curve is between two output signals, the strategy for choosing the sensor locations can be different compared with other metrics that might use modal data. As will be presented, the final sensor locations, which are the output of the present method, include some positions that fall in parts that are not moving and show no interest as a single location if someone considers the modes of the structure. Nevertheless, when these single sensors are used in combination with a sensor that is in other more interesting locations (in terms of modal information), the TF curve can provide very useful information for the structural condition of the structure. This capability creates the need to develop an OSP strategy that will be able to select the most promising sensor locations as a combination of sensors, rather than single sensors, in order to take full advantage of the TF information. While this strategy has been developed initially to be used with the current Damage Detection Framework, which is also incorporated, the final results (sensor locations) can be used with any other SHM methodology that considers the Transmittance Functions as a metric. 

In this new OSP strategy, a new layer is added on top of the Damage Detection Framework. The possible sensor locations are preselected in the FE model which corresponds to the examined structure. In the whole procedure, experimental measurements from the structure are not needed, although measurements from the healthy structure would be helpful to assure that the FE model can describe the dynamic response of the structure with accuracy. The whole evaluation will be done with simulated damage scenarios where the critical parts of the structure need to be selected. These parts can include all parts of the structure or only a small number of parts that might be considered as important, are prone to damage, or are in places that are difficult to inspect and where the researcher/engineer wants to make sure that damage will be detected. This new layer will create different scenarios with all the possible combinations of sensors. In each scenario, only the selected combination of sensors will be active. Then, the Damage Detection Framework will be called to find the simulated damage by having available only information from the active sensors. When all combinations and damage cases are evaluated, the final data are processed. The sensor combinations where the framework has found the simulated damage accurately are considered successful, and for these successful combinations their sensitivity is calculated. The final result is a list of sensor combinations that are able to find damage sorted from the highest to lowest sensitivity. 

The novelty of the proposed methodology lies in the fact that a combination of sensors is considered, which is crucial when the TF metric is used, compared with previous methods that select only single sensors in order to acquire the available information. The results that are presented show the difference between considering single sensors and a combination of sensors for damage detection purposes. The final task is to find the combinations that maximize the sensitivity of the experiment in structural damage and group the best combinations. Single sensor optimization has been the default practice in the literature, and no other methodology was found that takes into account combination of sensors to maximize the sensitivity of the TF metric. Single sensor methodologies can still be applied with the TF metric, but there is a chance that the final sensitivity of the experiment will be lowered at the end. If we consider two sensors that are affected by similar modes, the TF curve between them might not be sensitive to the damage as the response of the two sensors might be affected in a similar way. Additionally, the results show that by using the presented methodology, some locations that offer no interest in response (or modal data) as individual locations, when they are used for the calculation of the TF combining other locations in the structure, present significant improvement in sensitivity. As the OSP methodology incorporates the Damage Detection Framework, it ensures that it is able to find the damage with the available information as input. Another aspect that is inherited from the framework itself is that it can be applied to FE models that consist of any finite element type. Besides the sensitivity of the TF to structural damage, as it is an output-only metric it simplifies the procedure of creating simulated damage scenarios, as the excitation of the simulated scenario and the inspection experimental measurements do not need to be the same. 

This work is presented as follows. Section 2 describes briefly the Damage Detection Framework along with the optimization algorithm, the calculation of the Transmittance Function, and the final objective function in order to make it clear for the reader/researcher. Section 3 includes the description of the additional layer that is implemented for the OSP methodology. Section 4 presents the application of this OSP strategy on an experimental truss structure consisting of CFRP beams while considering three damage scenarios in different parts. Finally, the conclusions are summarized in Section 5.

## 2. Damage Detection Framework

The Damage Detection Framework that is used in the current work has been presented by the authors in the past [22]. As the modeling error between the FE model affects the accuracy of most model-based methods, an extended version [47] has also been developed to address this issue and minimize its effect while improving accuracy. A short mention of the framework is included in this section to assist in the understanding of the whole procedure, whereas for a more detailed description, the reader/researcher is encouraged to refer also to this past work. 

### 2.1. Particle Swarm Optimization (PSO) Algorithm 

A metaheuristic optimization algorithm is used at the core of the Damage Detection Framework. In the current version of the framework, the Particle Swarm Optimization (PSO) algorithm was selected.

PSO is a population-based algorithm that belongs to the subarea of Swarm Intelligence in the Computational Intelligence category, and was introduced by Kennedy and Eberhart [41,42,43].

The initial swarm (population), $Pop=\{p_{1},p_{2},\ldots ,p_{n}\}$, is sampled randomly and consists of n number of particles, $p_{i}\in R^{k}$ for i=1,2,…,n, where k is the number of parameters to be optimized. Each particle $p_{i}$ has a position, $x_{i}^{t}$, and velocity vector, $v_{i}^{t}$, at the given time step t.

The velocities and positions of each particle are updated in every cycle based on the following rules:(1)$$v_{i}^{t+1}=\underset{\mathrm{Inertia}}{\underbrace{w\cdot v_{i}^{t}}}+\underset{\mathrm{Cognitive}}{\underbrace{c_{1}\cdot R1_{\left(i,i\right)}\cdot (pbest_{i}^{t}-x_{i}^{t})}}+\underset{\mathrm{Social}}{\underbrace{c_{2}\cdot R2_{\left(i,i\right)}\cdot (lbest_{i}^{t}-x_{i}^{t})}}$$
(2)$$x_{i}^{t+1}=x_{i}^{t}+v_{i}^{t+1}$$
where w is the inertia weight and $c_{1},c_{2}$ the acceleration coefficient, which are all defined prior to calculation. R1,R2 are two k×k diagonal matrices with diagonal elements sampled at each iteration from a uniform random distribution with values from 0 to 1. Furthermore, pbest is a vector containing the best parameter set values of the corresponding particle and lbest is a vector containing the best parameter set values from the neighborhood particles, N. The neighborhood particles, N, are a fraction of the total number of particles n [42]. As with most optimization algorithms, the goal is to find the set of parameters that correspond to the minimum value of the objective function, G.

### 2.2. Damage Detection Framework

Considering a real-world structure with its corresponding Finite Element model that can be described by the equation of motion [48]:(3)$$M\;\ddot{x}+C\;\dot{x}+\;K\;x=F$$
where F is the external excitation and $\ddot{x},\dot{x},x$ represent the acceleration, velocity, and displacement vectors, respectively. M,C,K are the mass, damping, and stiffness matrix, respectively. 

When damage occurs in the real-world structure, a new damaged FE model is created that will indicate the damaged area. Assuming that the damage will affect the mass and stiffness matrices, the goal is to find the appropriate dM (changes in the mass matrix) and dK (changes in the stiffness matrix) from Equation (3). This is achieved by inserting a damaged area into the FE model. This area is controlled by the optimization algorithm which changes the position and the material mechanical properties. As such, the damage is expressed as a local change in the stiffness and mass. The search domain of the optimization algorithm includes six parameters in total. The first two represent the percentage change of the Elastic Modulus and Density, $p_{E},p_{D}$, by using Equations (4) and (5).
(4)$${\vec{E}}_{dam}=p_{E}\cdot {\vec{E}}_{part}$$
(5)$$D_{dam}=p_{D}\cdot D_{part}$$
*E*, *D* are the Elastic Modulus and Density. Furthermore, the subscript *dam* indicates the material properties (modulus, density) of the damaged area, and *part* is the material properties of the part in which the damaged area is inserted.

The exact location of the damaged area which is inserted into the FE model is controlled by the remaining four parameters. The vector L(P,X,Y,Z) describes the location of the inserted area, where *P* represents the part of the FE Model in a multi-part structure and *X,Y,Z* corresponds to the local coordinates expressed as a fraction of the total dimensions of the specific part chosen by *P*.

The import, manipulation, and export of the FE Model were implemented in MATLAB. For the evaluation of the dynamic response of the FE model, the commercial solver MSC NASTRAN was selected.

### 2.3. Transmittance Function

The Transmittance Function (TF) [44] is expressed as the ratio of the Cross-Spectral Density (CSD), $S_{rs}$, over the Auto-Spectral Density (PSD), $S_{rr}$, between two vibration response signals calculated from Equation (6).
(6)$$T_{rs}(\omega )=\frac{S_{rs}(\omega )}{S_{rr}(\omega )}=\frac{{\ddot{x}}_{r}(\omega )\;{\ddot{x}}_{s}^{*}(\omega )}{{\ddot{x}}_{r}(\omega )\;{\ddot{x}}_{r}^{*}(\omega )}$$
Here, $\ddot{x}(\omega )$ is the Fourier transform of the acceleration signal with *ω* as the frequency. Furthermore, ${\ddot{x}}^{*}(\omega )$ is the complex conjugate of $\ddot{x}(\omega )$, and subscripts *r*, *s* denote the degrees of freedom of the structure. In this case, *r*, *s* will represent the signals from the accelerometer sensors that are selected in the structure. 

### 2.4. Objective Function 

The optimization task with the PSO algorithm is a single objective minimization procedure. The objective function is formulated by using the Pearson Correlation Coefficient (ρ), which takes values between the range of −1 and 1. A value of ρ=1 indicates that there is a perfect linear correlation between the two data sets, ρ=−1 indicates a negative linear correlation, while ρ=0 indicates that there is a nonlinear relationship but without providing any further details.

The final objective function, G in Equation (7), is the mean value of the errors (from the linear correlation, ρ=1) of the Pearson coefficients between the experimental measurements of the damaged structure, $TF^{DEXP}$, and the FE model, $TF^{FE}$, where *A* denotes the total number of Transmittance Functions used with *i* = 1, 2, …, *A*.
(7)$$Objective\;Function,\;\;\;\;G=\frac{1}{A}\sum_{i=1}^{A}\left[1-\rho \left(T_{i}^{DEXP},T_{i}^{FE}\right)\right]$$

The flow chart of the Damage Detection framework is presented in Figure 1.

## 3. Optimal Sensor Placement

As mentioned in Section 1, a common procedure to decide the location of the sensors is based on the modal information of the structure while different dynamic metrics are considered. The Damage Detection Framework of Section 2 is used within the OSP method. The metric that is incorporated in this framework is based on the Transmittance Function, which was described in Section 2.3. This metric is calculated by using signals from two different acceleration sensors. As such, the optimal placement can be different compared to a method that uses the modes of the structure. In the current case, the focus is to find the best combination of sensors across the structure. This is achieved by adding an extra layer on top to call different instances of the damage detection framework which evaluates a simulated damage scenario with different sensor combinations. 

Consider a healthy real-world structure and its corresponding FE model. The first step is to select the possible sensor locations that will be evaluated. These locations can be spread over different parts, or locations in the same part, depending on the examined structure. From the selected locations, all the possible sensor combinations will be created. In this combination list, there are cases that include one, two, or three active sensors, up to the limit of all possible selected sensors. As such, it is important to set a limit on the active sensors to restrict the list of cases to be evaluated. The minimum number of active sensors needs to be two to calculate the Transmittance Function, and while two sensors can be the maximum limit, more combinations can also be included if needed. The next step is to create a virtual damage scenario on the part (or location in case of a single-part structure) and extract the acceleration measurements from all the selected sensor locations. These measurements will be used to replace the experimental measurements of the damaged structure inside the Damage Detection Framework. For each sensor combination, the framework will be called to find the damaged location by taking into account only the acceleration signals from the active sensors. When all cases with different sensor combinations are evaluated, the data are collected. These data include the best solutions (damage location) that the framework found for each combination. The data from all evaluated combinations are cleaned, and only the combinations where the final indicated damage location from the framework is correct are kept; the rest of the cases can be disregarded. 

The final step is to sort the correct combinations based on their sensitivity. As explained in the original presentation of the Damage Detection Framework [22,47], the result of the framework is considered in a binary form: either it is correct, and the real damage of the structure is included in the indicated damage location from the framework, or it is false. As such, the sensitivity of each combination is calculated based on the relative value of the objective function (ROF), Equation (8). The ROF is calculated from the value of the objective function, Equation (7), before and after the execution of the Damage Detection Framework for the specific sensor combination.
(8)$$Relative\;Objective\;Function\;\;ROF=G_{Healthy\;FE-Sim.Damage\;Case}-G_{Damaged\;FE-Sim.Damage\;Case}$$

$G_{Healthy\;FE-Sim.Damage\;Case}$: The value of Equation (7) calculated between the acceleration signals of the healthy FE model and the FE model of the simulated damage case that is being evaluated.

$G_{Damaged\;FE-Sim.\;Damage\;Case}$: The value of Equation (7) between the final damaged model which is the output of the Damage Detection Framework and the FE model of the simulated damage case that is being evaluated. This is the actual output objective function of the framework which has been identified as the best value. 

The first term of Equation (8) shows the influence of the simulated damage on the healthy structure. The second term is the final objective function, which is the correlation of the final damaged model with the simulated damage scenario. Higher ROF values indicate that the sensor combinations are more sensitive for this damage case and are ranked higher on the list.

The value of the final objective function from the framework, the second term of Equation (8), is expected to be small as the optimization algorithm has minimized this value. Nevertheless, the final sorting of the combinations based on the ROF value can play a role, depending on the structure and the selection of the possible sensor locations. The final output will be a list of sensor combinations that are able to find the location of the damage. This list will be sorted based on the ROF value and will correspond to one damage scenario. The final decision on the sensor location will be based on this list, but possibly also on evaluations of other damage scenarios. As such, the researcher/engineer might not choose the first combination in this list, with the best ROF value, but a combination that is near the top of the list while also satisfying other criteria. 

The flow chart of the sensor placement procedure is presented in Figure 2.

## 4. Validation

### 4.1. Experimental Setup and FE Model

The presented procedure is applied to a truss structure that consists of Carbon Fiber Reinforced Composite (CFRP) tubes and aluminum connectors. The aluminum connectors are glued to the ends of the CFRP tubes; these connectors are then bolted onto intermediate aluminum parts. The complete structure is permanently clamped on a steel base which is attached to a concrete wall. The excitation of the structure is produced by an electrodynamic shaker connected at one corner of the truss via a stinger rod, and random excitation is applied. 

The FE model of the structure consists of shell elements for the composite material and solid elements for the aluminum connectors, the intermediate parts, and the glue. The commercial solver MSC Nastran was used and a modal frequency response analysis was selected. More details about this structure and the FE model can be found in a previous article of the authors [47]. Based on the results of the frequency response analysis, the acceleration output signals are used for the calculation of the Transmittance Functions (TF), Equation (6). 

Figure 3 shows the complete experimental setup, while in Figure 4 the FE model is presented.

Fourteen possible sensor locations were selected, of which ten are located in the center of the CFRP tubes and four on the intermediate aluminum parts as presented in Figure 5. The sensors were named based on the ID number for the CFRP tubes, shown in Figure 6, or based on their location for the sensors placed in the intermediate parts (with respect to Figure 5).

Three different simulated damage cases (D1, D3, and D7) were evaluated for this structure, located in the CFRP tubes with ID 1, 3, and 7. Figure 7 shows all three of the damage scenarios in the FE model. The simulated damage scenarios were created on the diagonal tubes 1, 3, and 7 at a distance of 540 mm from the base end of the tube. The damage was simulated by reducing the stiffness of the damaged area, indicated with red color in Figure 7, by 10%.

The evaluation of the optimal sensor locations does not require experimental measurements from a damaged structure. Nevertheless, similar damage was created on the structure in order to compare the sensitivity of the sensor combinations between the real-world structure (healthy and damaged) and the FE model (healthy FE model and simulated damage scenario). Figure 8 shows the damage that was created on one of the diagonal tubes in the CFRP material. The tube was placed in a tree-point bending scenario in a compression machine. Rubber material was placed under the two points that hold the tube, thus preventing unwanted damage in these regions. The resulting damage is a local reduction of stiffness as multiple local cracks were created in the composite material. As the damage cases each include a tube with the same characteristics, this damaged part can be mounted in all three different positions on the specific structure. It must be noted that the stiffness reduction percentage of the real damage is not known, and it should not be compared quantitively with the simulated damage reduction. The comparison between the healthy-damaged structure and the healthy-damaged FE model will show only that the sensitivity pattern of the sensor combination is similar.

From the sensor locations that were selected in Figure 5, both the *Z* and *Y*-axis signals are taken into account in every sensor combination, while the *X*-axis is disregarded as the structure is clamped on the wall. Accordingly, in the experimental structure, the fourteen sensors are triaxial accelerometers where the *X*-axis is again disregarded.

### 4.2. Optimal Sensor Placement

The first investigation is damage case D1 which is located on the CFRP tube with ID 1. Before the execution of the methodology, a comparison was made between the simulated and experimental data. The comparison was made by calculating Equation (7). For the experimental data, this equation is calculated between the real-world healthy structure and the damaged structure each time by taking into account only the acceleration signals of the active sensor combination, shown in Figure 9. Accordingly, for the FE model, the same equation is used between the healthy FE model and the simulated damage case that was created, shown in Figure 10. The comparison of the two figures, Figure 9 and Figure 10, reveals that the pattern is the same. However, the absolute values between the two graphs have differences. This should not be taken into account as the experimental data are not used for the evaluation of the sensor locations and, as was noted before, the percentage of stiffness reduction in the real damage is unknown. The important information is that the sensitivity of each combination follows the same pattern in the real structure and the FE model. 

The simulated damage case and the healthy FE model were used as inputs for the methodology that was presented in Section 3. Once all the sensor combinations had been evaluated, all data were gathered and processed. It must be noted that for all the damage cases in the present work, only combinations with two sensors were evaluated for this structure. Figure 11 shows the ROF values, Equation (8), for all the sensor combinations. Similarly, the same data are presented in Figure 12 with the distinction that only the successful sensor combinations are included. A sensor combination is considered successful when the damage detection framework used this combination and was able to find the damage location. As such, the remaining sensor combinations are disregarded for this damage case.

The results in Figure 12 show some interesting combinations. The most sensitive combination for damage in this part seems to be “1-BLL” while “1–7”, “6–8”, “6–7”, and “2–4” follow. It was expected that sensor “1” would be at the top of the list, as it is located on the same part as the damage, whereas the location of “BLL” is placed upon a part that is fixed. Other similar combinations include “7-BLL” and “5-BUR”, although with a lower ROF value. Specifically, for locations such as “BLL” and “BUR” that are almost fixed, methods that rely directly or indirectly on the modes of the structure would probably disregard these locations. By highlighting combinations such as these, the presented methodology shows its value when the Transmittance Function is used as a metric for damage detection purposes.

Indicatively, the output of the Damage Detection Framework is presented in Figure 13 when it was called to evaluate damage case D1 with the active sensor combination of “1-BLL”. Similar results were obtained with all the combinations that were successful and are included in Figure 12.

The same procedure was followed for the two other damage cases, D3 and D7, which are located in the CFRP tube with ID 3 and 7, respectively.

Regarding case D3, Figure 14 and Figure 15 show the values of the objective function, Equation (7), between the healthy and damaged states of the experimental structure and the FE model, respectively, when it is calculated separately for each combination. It is obvious that both images again follow the same pattern as was the case for D1. Figure 16 and Figure 17 present the ROF values of all the evaluated sensor combinations, and the successful sensor combinations, respectively. From the successful combinations, “2–3” is placed at the top of the list, where location “3” might be expected. Following are combinations “9–11”, “6–8”, and a combination that includes a location on a part of the structure that is fixed, “7-BLL”.

Regarding case D7, Figure 18 and Figure 19 show the values of the objective function, Equation (7), between the healthy and damaged states of the experimental structure and the FE model accordingly. Figure 20 and Figure 21 present the ROF values of all the evaluated sensor combinations, and the successful sensor combinations, respectively. Sensor combinations “7-BLL” and “6–8” are placed at the top of the list while combinations “1–7”, “6–7”, “6-FUL”, and “1–6” follow.

As a summary, Table 1 presents the top ten sensor combinations for all three damage cases. The list is sorted in descending order based on the ROF values. As noted, the final results show some interesting locations such as “BLL”, which is placed in a part that is fixed. Furthermore, the final list highlights the importance of some sensor combinations between different parts that do not include the damaged part.

In the investigation that was performed in the present study, three parts were chosen as critical parts. This means that the engineer/researcher can choose the final sensor locations by combining these results to ensure that damage that could occur in any of these parts can be found with the selected sensor locations. Because of the geometry of the structure, which is a small truss, more than one sensor for each part was found to be unnecessary. It is logical to say that more damage cases in different parts should also be investigated, and the procedure would be identical to the one presented in the current section. Since the OSP methodology relies on simulated damage cases, it is possible to investigate as many damage cases as seem sufficient, simple or complex.

The reader might reasonably question the performance of this OSP method in cases where damage to the structure has not been investigated as a virtual damage case. The purpose of the virtual damage cases is not necessarily to investigate all the possible damage that could appear in the structure. They are used to evaluate the sensitivity of the sensor combination in this specific damaged part of the structure, while in parallel ensuring that if damage occurs in this part it can be found. For example, in the truss structure investigated in this section, the engineer/researcher does not need to investigate more than one case of damage in the same tube. Furthermore, the choice of virtual damage cases can be made by the researcher/engineer either by selecting only critical parts or all parts of the structure. 

## 5. Conclusions

The presented methodology can evaluate the optimal placement of sensors based on a group of preselected possible locations by taking into account simulated damage cases. A damage detection algorithm is used as a tool within the new method. The metric that is connected with the dynamic response of the structure and is used for the evaluation is the Transmittance Function. As this metric is calculated between two acceleration signals, the final result of the methodology is a combination of sensor locations instead of individual sensors. The application of the methodology to the truss structure shows some interesting features. The results for each of the three presented cases, but also the whole set, include locations that could be disregarded with other methods as they are based on non-moving parts and normally attract no interest in vibration-based methods. This is an effect of using the Transmittance Function as a metric. Furthermore, by applying the damage detection framework for each sensor combination, the list is filtered regardless of their sensitivity. A combination may present a high sensitivity, but if it is not able to find the damage location it should not be taken into account. The sensitivity is measured using the ROF value, which is a relative metric based on the objection function instead of an absolute value of the same function. While on the current structure the most valuable sensor combinations were relatively easy to identify, in more complex structures, or if more than one sensor location is selected in the same part, sorting based on the ROF value can show its advantage. It must be noted that the selection of the possible sensor location must be made with caution. The number of possible sensor combinations grows rapidly with every newly selected location, so the computational cost will grow accordingly. The procedure of applying the methodology in larger and/or more complex structures is identical with the one presented in the current work, assuming that an accurate FE model of the structure exists or can be developed. The only difference would be the number of different simulated damage cases that should be investigated, and possibly the number of possible sensor locations (and combinations), which, as mentioned before, could increase the total computational cost. Nevertheless, the fact that this methodology incorporates a tested damage detection methodology gives higher confidence for sensor placement in a structure for damage detection purposes.

## Figures and Tables

**Figure 1 sensors-24-01608-f001:**
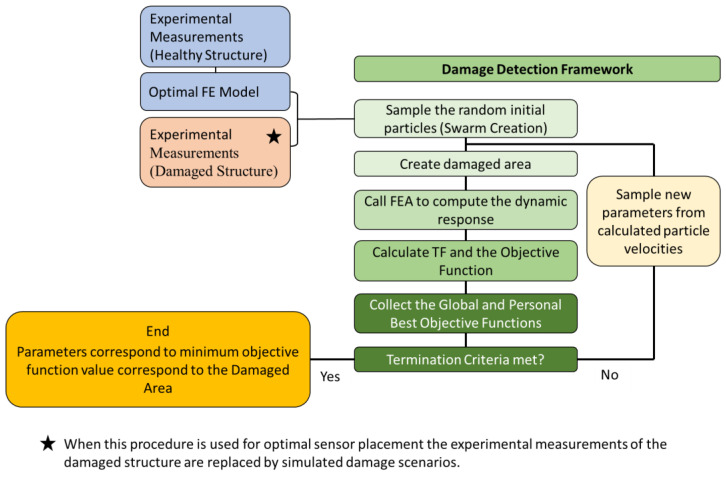
Flow chart of the Damage Detection Framework [22].

**Figure 2 sensors-24-01608-f002:**
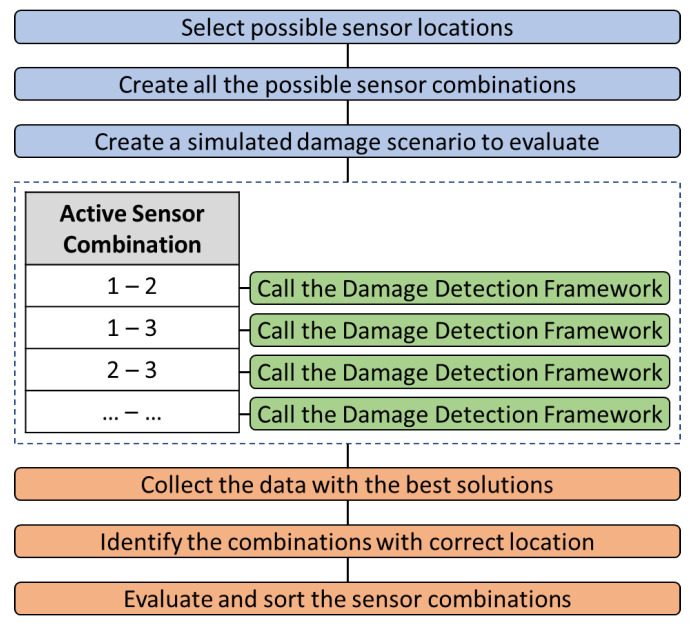
Flow chart of the optimal sensor placement procedure.

**Figure 3 sensors-24-01608-f003:**
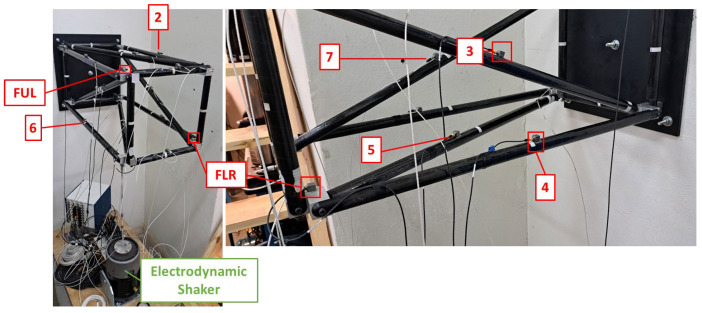
Experimental setup of the CFRP truss [47].

**Figure 4 sensors-24-01608-f004:**
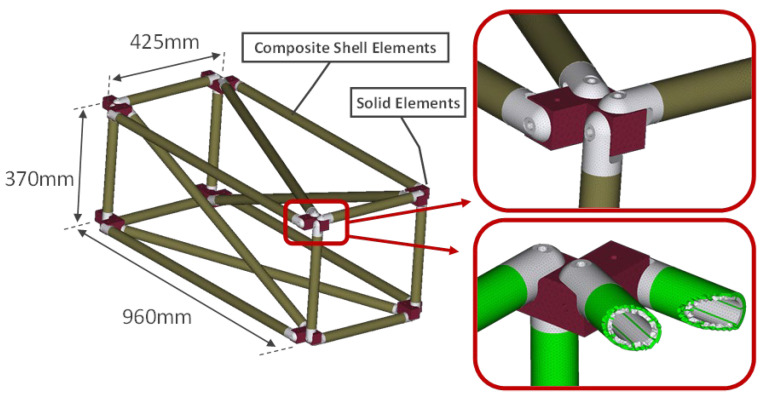
Finite Element model of the truss [47].

**Figure 5 sensors-24-01608-f005:**
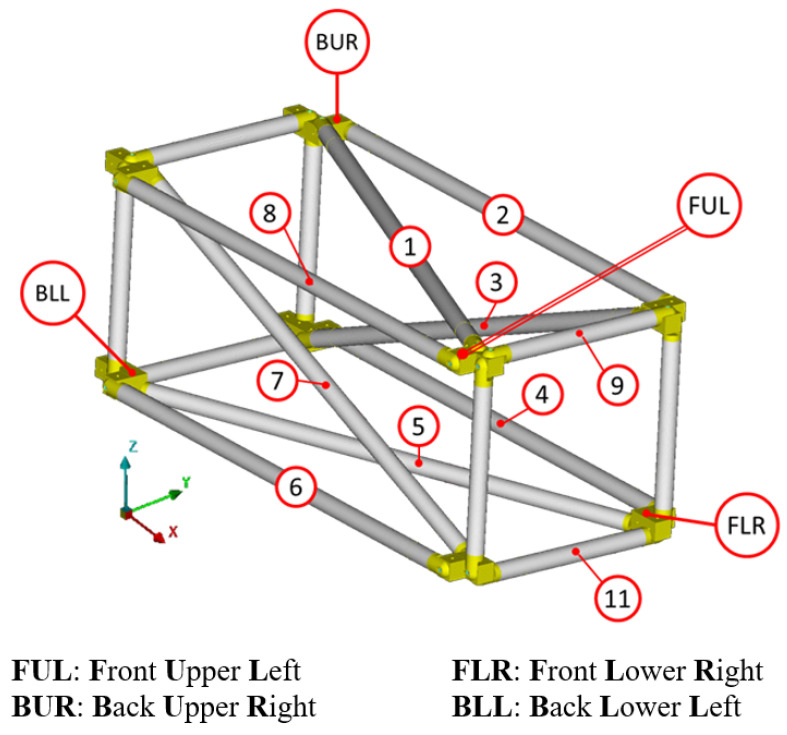
Selected sensor locations.

**Figure 6 sensors-24-01608-f006:**
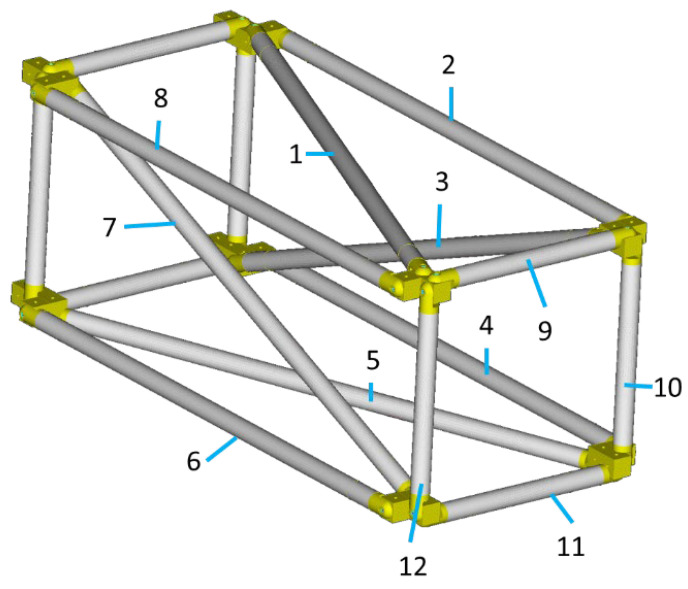
CFRP tube identification numbers.

**Figure 7 sensors-24-01608-f007:**
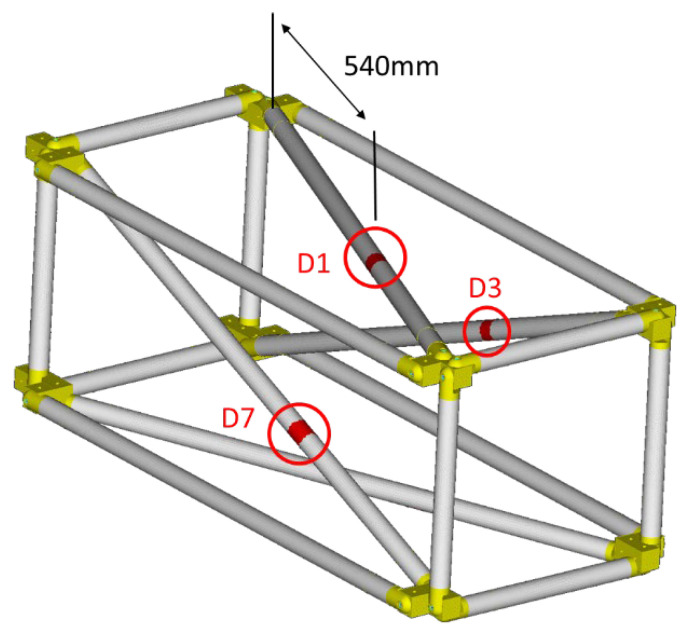
Damage cases.

**Figure 8 sensors-24-01608-f008:**
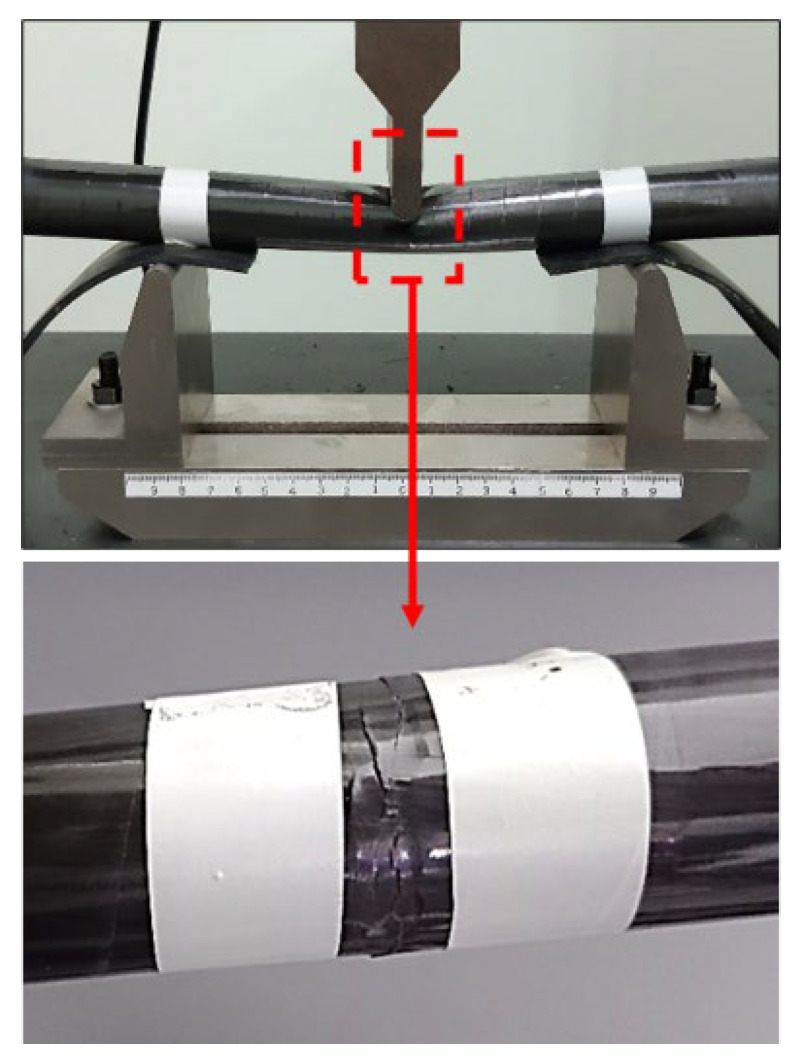
Damage creation on the experimental structure.

**Figure 9 sensors-24-01608-f009:**
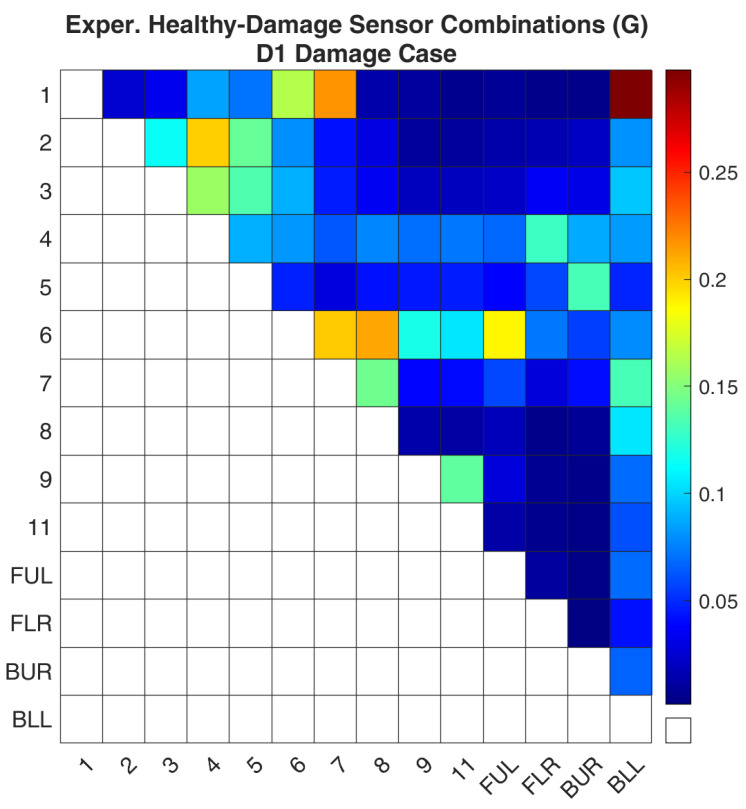
Experimental Healthy-Damaged values of the objective function (G) for the D1 damage case.

**Figure 10 sensors-24-01608-f010:**
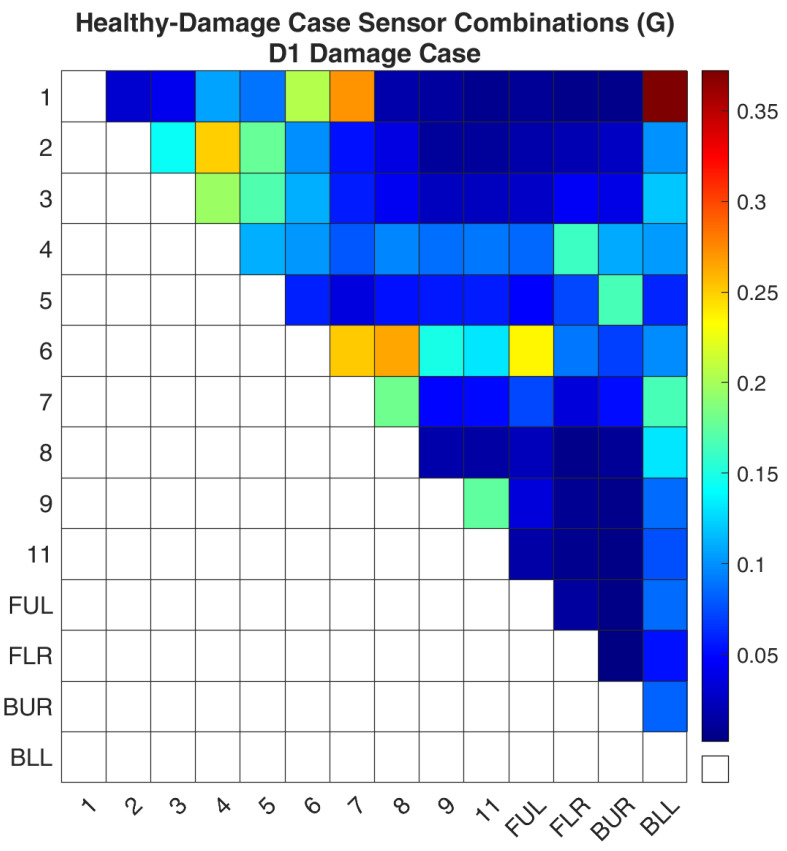
FE model-Simulated damage values of the objective function (G) for the D1 damage case.

**Figure 11 sensors-24-01608-f011:**
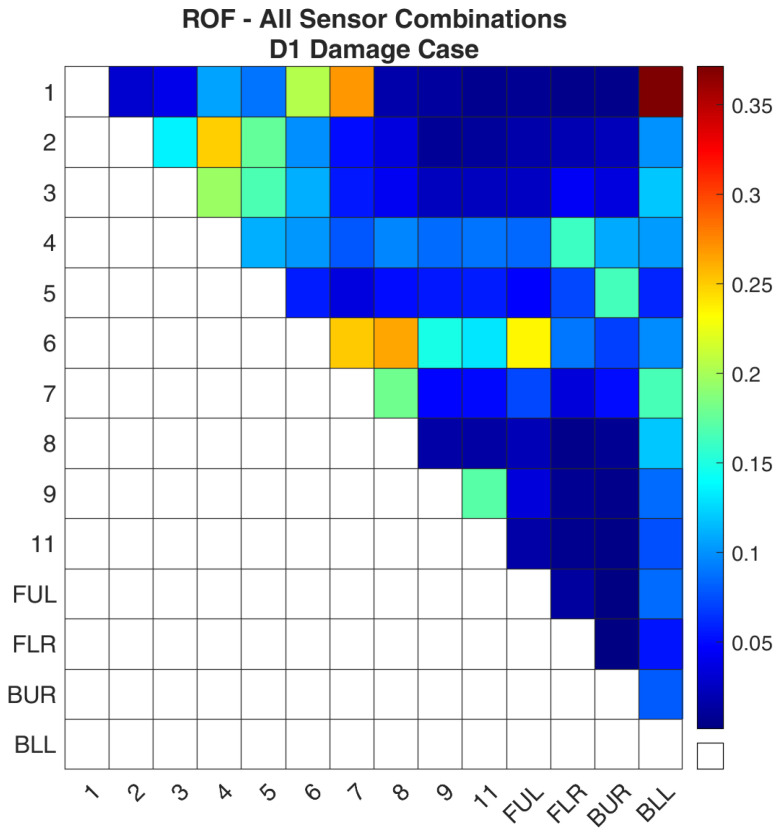
ROF of all the sensor combinations for the D1 damage case in the FE Model.

**Figure 12 sensors-24-01608-f012:**
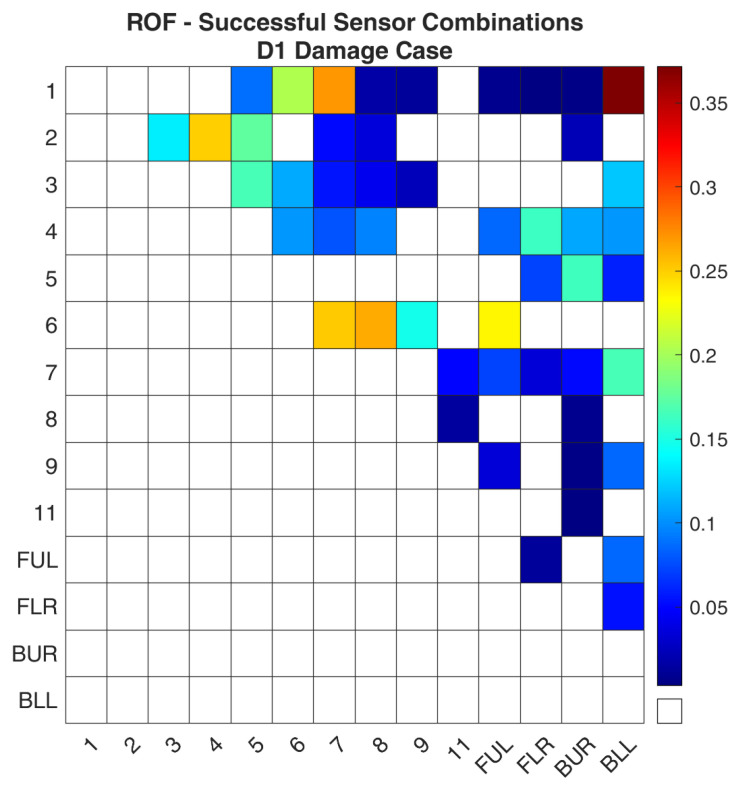
ROF of the sensor combinations that found the damage location of the D1 damage case in the FE Model.

**Figure 13 sensors-24-01608-f013:**
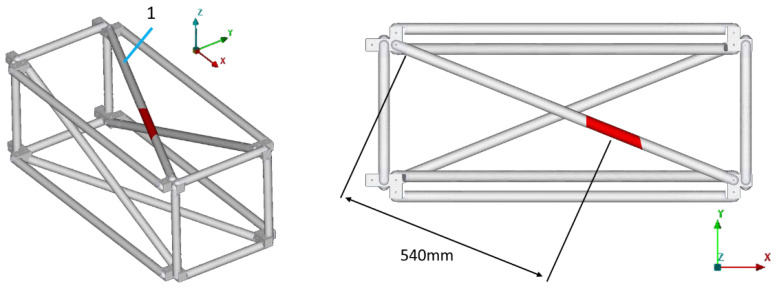
Damaged FE model of the D1 damage case, the output of the Damage Detection Framework, using sensor combination 1-BLL.

**Figure 14 sensors-24-01608-f014:**
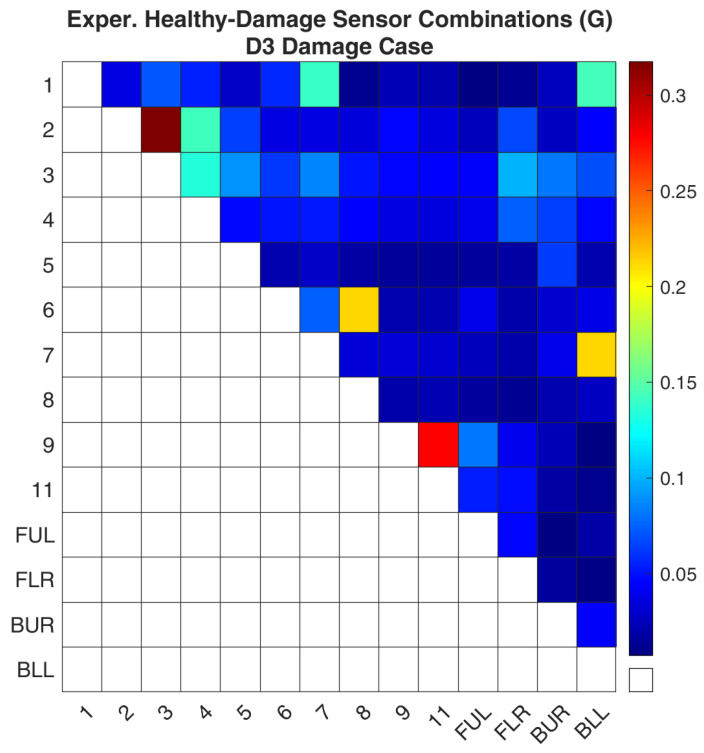
Experimental Healthy-Damaged values of the objective function (G) for the D3 damage case.

**Figure 15 sensors-24-01608-f015:**
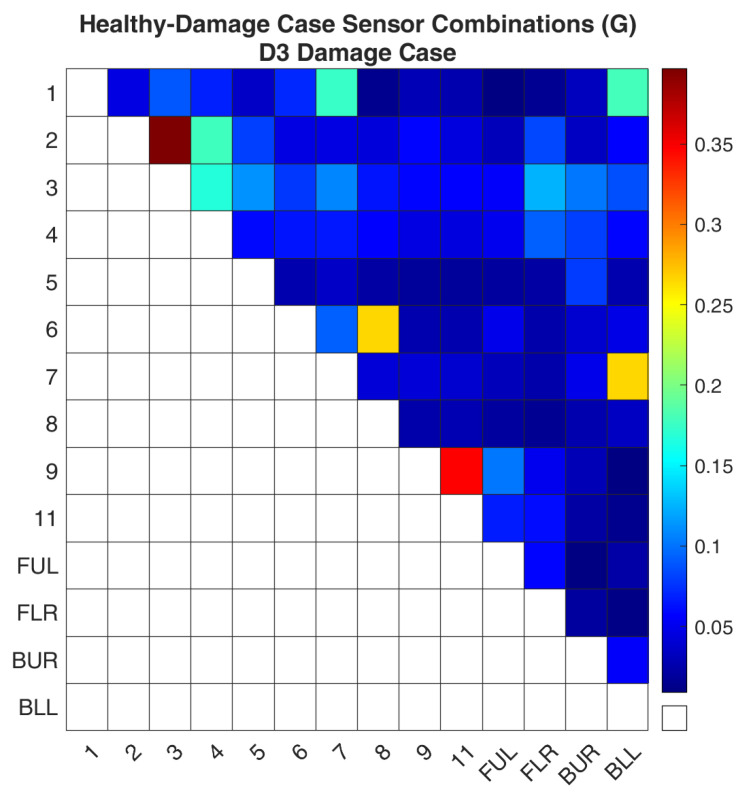
FE model-Simulated damage values of the objective function (G) for the D3 damage case.

**Figure 16 sensors-24-01608-f016:**
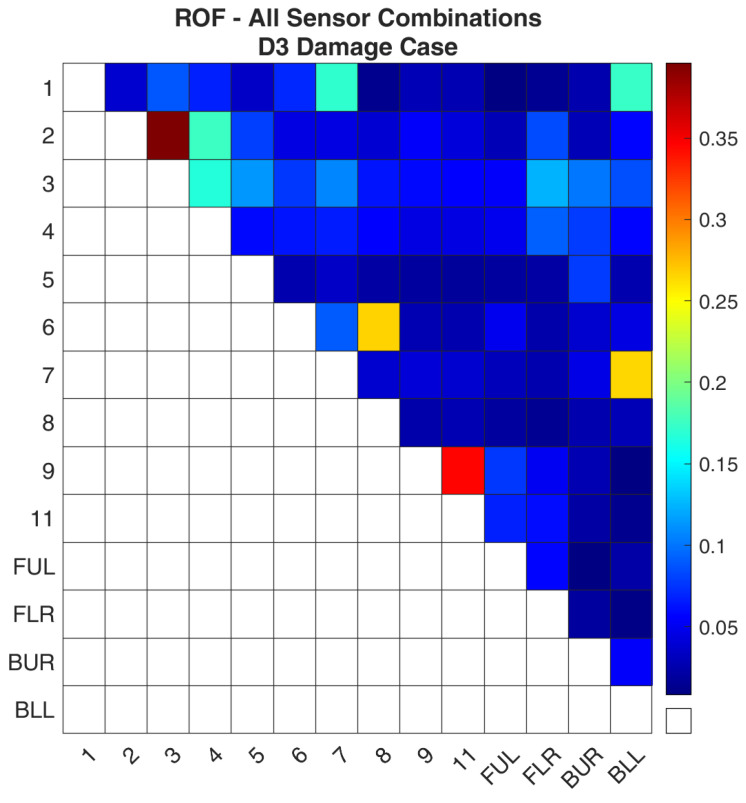
ROF of all the sensor combinations for the D3 damage case in the FE Model.

**Figure 17 sensors-24-01608-f017:**
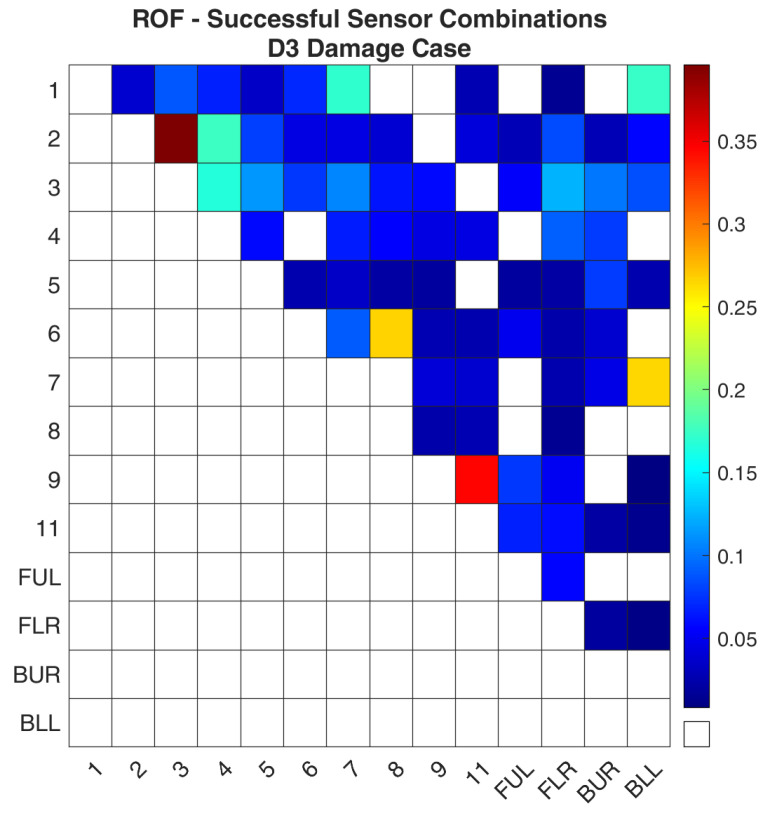
ROF of the sensor combinations that found the damage location of the D3 damage case in the FE Model.

**Figure 18 sensors-24-01608-f018:**
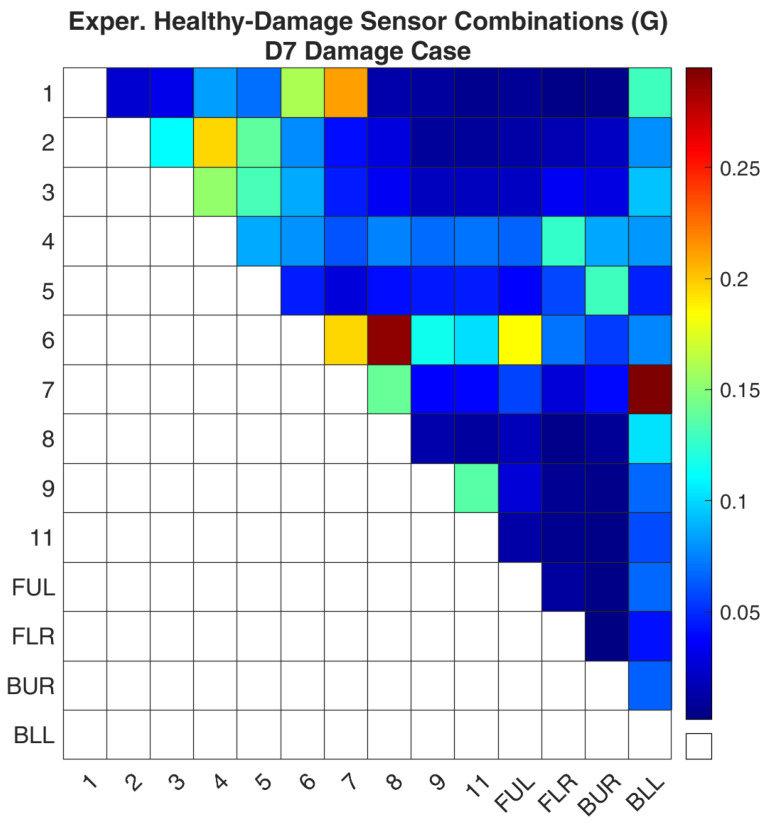
Experimental Healthy-Damaged values of the objective function (G) for the D7 damage case.

**Figure 19 sensors-24-01608-f019:**
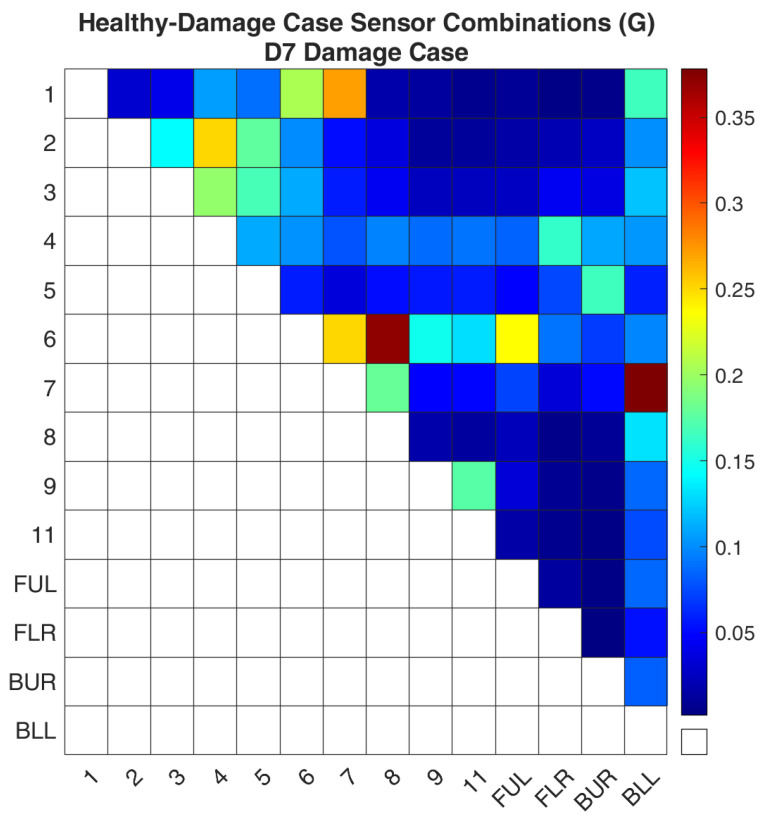
FE model-Simulated damage values of the objective function (G) for the D7 damage case.

**Figure 20 sensors-24-01608-f020:**
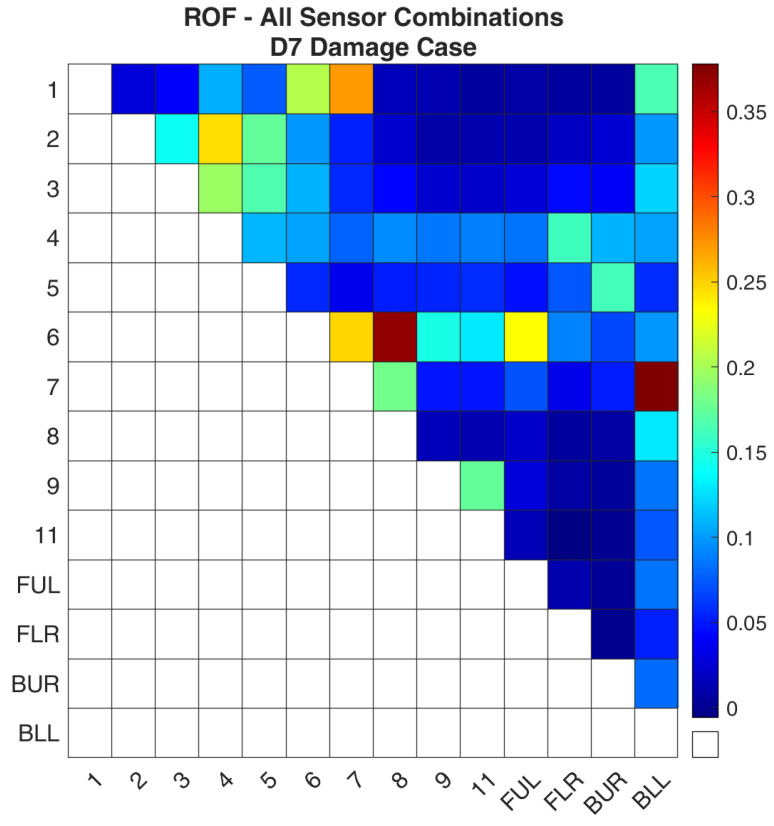
ROF of all the sensor combinations for the D7 damage case in the FE Model.

**Figure 21 sensors-24-01608-f021:**
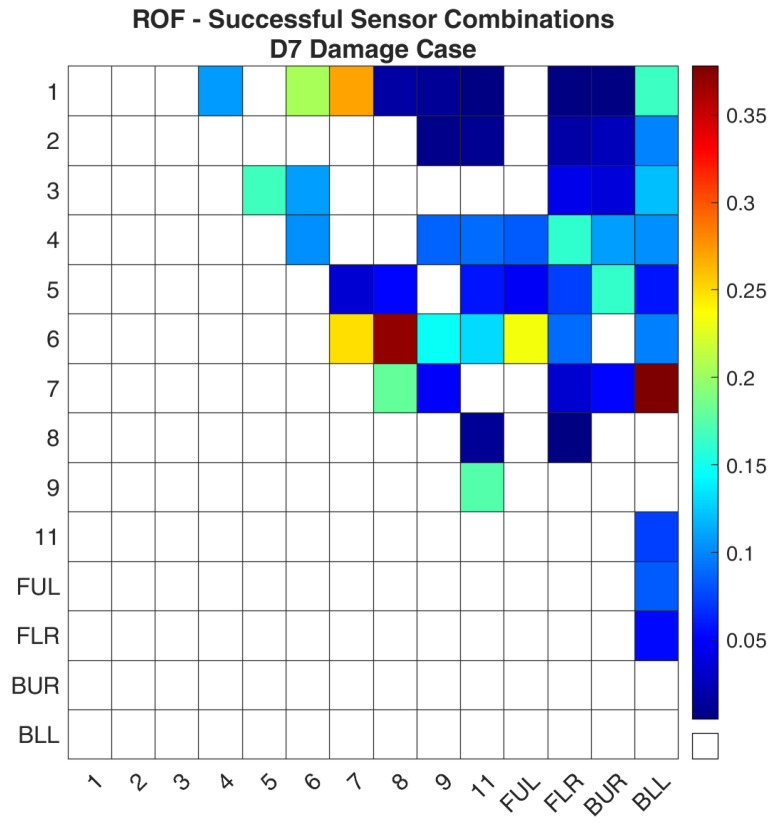
ROF of the sensor combinations that found the damage location of the D7 damage case in the FE Model.

**Table 1 sensors-24-01608-t001:** The top 10 successful sensor combinations for all three damage cases, sorted based on ROF value.

	D1			D3			D7		
A/N	Sensor Combination		ROF	Sensor Combination		ROF	Sensor Combination		ROF
1	1	BLL	0.371	2	3	0.395	7	BLL	0.377
2	1	7	0.270	9	11	0.344	6	8	0.369
3	6	8	0.263	6	8	0.265	1	7	0.271
4	6	7	0.250	7	BLL	0.264	6	7	0.248
5	2	4	0.249	2	4	0.175	6	FUL	0.233
6	6	FUL	0.234	1	BLL	0.173	1	6	0.205
7	1	6	0.203	1	7	0.171	7	8	0.180
8	2	5	0.175	3	4	0.166	9	11	0.174
9	3	5	0.166	3	FLR	0.124	3	5	0.167
10	7	BLL	0.165	3	5	0.112	1	BLL	0.165

## Data Availability

Dataset available on request from the authors.

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
