# Peer review of "Optimal Sensor Placement for Vibration-Based Damage Localization Using the Transmittance Function"

_sensors, 2024, doi:10.3390/s24051608_

Round 1

Reviewer 1 Report

Comments and Suggestions for Authors

The paper discusses a method for optimal sensor placement based on the transmittance functions. The paper fits well the topics of the Journal, however, it is opinion of this referee that the manuscript can be considered for publication after the Authors have considered the following comments.

1.      To improve the readability, Authors must cite in the text all the symbols in eq (4) and (5).

2.      Section 3.1 the most important geometrical features of the elements that compose the specimen, must be inserted for example in the figure 3 or 4, in order to facilitate the comprehension. Moreover, some data must be introduced on the force acting on the specimen and on the installed sensors.

3.      The figure 8 shows that the element has been significantly damaged, and a deflection occurs on the damaged tube. The Authors may insert a sentence on the efficacy of the procedure in cases with a lower damage level.

4.      To provide advantages and limits of the chosen approach with respect to other methods, a description of the available methods for detection of local damage must be discussed. Following some references to other approaches available in literature

A variance-based approach for the detection and localization of cracks in a beam. Structures Volume 44, Pages 1261 – 1277, October 2022

Experimental structural damage localization in beam structure using spatial continuous wavelet transform and mode shape curvature methods. (2017) Measurement: Journal of the International Measurement Confederation, 102, pp. 253-270

Modal identification of damaged frames  (2016) Structural Control and Health Monitoring, 23 (1), pp. 82-102.

5.      The Authors must introduce some information regarding the forces acting on the specimen, the amplitude of the accelerations and the accuracy of the sensors. Finally, some considerations must be introduced on the efficacy of the proposed method when the damage level is lower than that examined in the paper and/or the geometrical features of the whole structure are different, and thus, a lower amplitude of vibrations are expected. All the experimental data are affected by noise; thus, the Authors must specify which is the accuracy of the method when for example the damage level is not high and, thus, the influence of noise on the signals may be significant.

6.      The Authors must specify the model adopted to describe the damage in the FE model and the assumption for the definition of the FE model of undamaged structure. Additional details must be included regarding for example, the real force applied to the specimen and the force adopted in the FE model, the effect of noise on the experimentally acquired signal, etc.

7.      The captions of figures 11 and 12 must specify if they refer to FE model simulated or experimental data.

8.      The Authors specify “It is logical to say that more damage cases in different parts can also be investigated, and the procedure will be identical to the one presented in the current section.”, additional data can be added to support this sentence, for example utilizing simulated data.

Comments on the Quality of English Language

1.      Please, revise typographical errors throughout the manuscript.

Author Response

Modifications in Response to Comments of Reviewer # 1

Answers are provided per comment

We would like to thank the reviewer for devoting his/her time to examine and study our paper and for providing constructive criticism and helpful suggestions. In preparing the revised manuscript we considered all comments, requests and suggestions, as explained next in detail. All changes in the text are highlighted by yellow color in the revised manuscript.

  1. To improve the readability, Authors must cite in the text all the symbols in eq (4) and (5).

    Thank you for your comment. The symbols of these equations are now cited in the text and highlighted in yellow.

  1. Section 3.1 the most important geometrical features of the elements that compose the specimen, must be inserted for example in the figure 3 or 4, in order to facilitate the comprehension. Moreover, some data must be introduced on the force acting on the specimen and on the installed sensors.

Thank you for your comment. More details about the model, structure and experimental procedure are provided in the reference that is mentioned in Section 4.1 second paragraph (Reference 47).

The intention of the authors was to avoid duplicating all the details that already exist in the previous work from the authors where the Damage Detection Framework is applied. In order to make it easier for the reader the main details of the structure, model and experimental procedure is mentioned but for more in depth details the reader can refer to the Reference that has been provided.

In that way the reader can focus more clearly in the discussed topic that is the Optimal Sensor Placement methodology that is presented.

If the reviewer requires it the authors can include all these details but these details will be the same as it is in the provided Reference.

[47] I. Zacharakis and D. Giagopoulos, "Model-Based Damage Localization Using the Particle Swarm Optimization Algorithm and Dynamic Time Wrapping for Pattern Recreation," Sensors, vol. 23, no. 2, 2023, doi: https://doi.org/10.3390/s23020591.

  1. The figure 8 shows that the element has been significantly damaged, and a deflection occurs on the damaged tube. The Authors may insert a sentence on the efficacy of the procedure in cases with a lower damage level.

Thank you for your comment. As it is indicated in Page 10 on the second paragraph (starting line 312) the experimental procedure is not required for the evaluation of the Optimal Sensor Placement.
Nevertheless, it is presented in order to assist in the comprehension and also validate that the sensitivity pattern of all sensors combinations are equivalent between the experiment and the FE model.

The experimental damage might appear as significant in the image but in reality, the part still maintaining a big percentage of stiffness. The deflection that appears in the images is due to a large number of micro cracks and possibly some areas that enter the plasticity region. The complete structure is very stiff due to the geometry and the materials that are used. As a result, the total effect of the damage is small.

When performing the Optimal Sensor Placement procedure the reduced stiffness is only 10 % in a very small region of the tube. This is mentioned in Page 10 at the end of the first paragraph (line 311) and in Figure 7.

The damage level is relative as it may be introduced a small region with a high stiffness reduction or in large area with a big stiffness reduction. The position of this damage will also affect in such geometries with multiple parts. As such it is difficult to comment with certainty without confusing the reader. As this is a procedure that is performed with simulated data only, the level of damage can be lower even from the 10 % that is introduced currently.

The authors believe that first the results will be similar since the procedure relies only to simulated data and furthermore, specific mention in the percentage of damage might confuse the reader/ researched that could potentially try to apply the method in a different structure.

  1. To provide advantages and limits of the chosen approach with respect to other methods, a description of the available methods for detection of local damage must be discussed. Following some references to other approaches available in literature
  2. A variance-based approach for the detection and localization of cracks in a beam. Structures Volume 44, Pages 1261 – 1277, October 2022
  3. Experimental structural damage localization in beam structure using spatial continuous wavelet transform and mode shape curvature methods. (2017) Measurement: Journal of the International Measurement Confederation, 102, pp. 253-270
  4. Modal identification of damaged frames  (2016) Structural Control and Health Monitoring, 23 (1), pp. 82-102.

Thank you for your comment and for providing relative literature. The references have been added along with additional research in the revised manuscript.

  1. The Authors must introduce some information regarding the forces acting on the specimen, the amplitude of the accelerations and the accuracy of the sensors. Finally, some considerations must be introduced on the efficacy of the proposed method when the damage level is lower than that examined in the paper and/or the geometrical features of the whole structure are different, and thus, a lower amplitude of vibrations are expected. All the experimental data are affected by noise; thus, the Authors must specify which is the accuracy of the method when for example the damage level is not high and, thus, the influence of noise on the signals may be significant.

Thank you for your comment.

This comment was been partially answered in the previous comments of the reviewer.

More specifically:

Regarding the acting forces on the structure, the geometrical features of the specimen, acceleration amplitude the reviewer is kindly requested to refer in the answer of his comment #2.

Regarding the level of damage the author is kindly requested to refer in the answer of his comment #3.

Furthermore, the Optimal Sensor Placement (OSP) method that is presented required only simulation data. The experimental measurements are not taken into account. In the present work the authors investigated the experimental behavior of the structure in order to verify that the sensitivity of sensor combination is following the same pattern between the physical structure and the FE model. This pattern can be found in Figure 9-10 , 14-15 and 18-19. The above comment is mentioned inside the manuscript in Section 4.1 ( below Figures 5 and 6). As such there is no noise level that needs to be examined in order to execute the OSP method.

  1. The Authors must specify the model adopted to describe the damage in the FE model and the assumption for the definition of the FE model of undamaged structure. Additional details must be included regarding for example, the real force applied to the specimen and the force adopted in the FE model, the effect of noise on the experimentally acquired signal, etc.

Thank you for your comment.

This comment was been partially answered in the previous commens of the reviewer.

More specifically:

Regarding the experimental details such as the force applied in the structure and FE model the reviewer is kindly requested to refer in the answer of his comment #2.

Regarding the effect for the application of the presented OSP methodology the reviewer is kindly requested to refer in the answer of his comment #5.

Regarding the model that was adopted to describe the damage in the part. The damage is simulated with a simple local stiffness reduction of 10%. This is referred in Section 4.1 lines 309-310. As it is mentioned in the comment #5 and inside the manuscript the intention of performing the experimental procedure is to compare the pattern of the sensitivity between the sensor combinations. As such the exact representation of damage in the FE model is not required. When the OSP method will be applied there is no need for experimental measurements with damage and the virtual scenarios can be simulated with any damage model (simple or complex) that the researcher might choose.  

  1. The captions of figures 11 and 12 must specify if they refer to FE model simulated or experimental data.

Thank you for your comment. The captions of Figures 11 and 12 have been revised along with Figures 16, 17, 20 and 21.

  1. The Authors specify “It is logical to say that more damage cases in different parts can also be investigated, and the procedure will be identical to the one presented in the current section.”, additional data can be added to support this sentence, for example utilizing simulated data.

Thank you for your comment. As it was mentioned in the reviewers comment # 5,#6 and also inside the manuscript in Section 4.1, lines 312-316, the method is based on simulated data and this is the reason that this sentence has been included.

A comment has been added in the manuscript in order to highlight this.

Comments on the Quality of English Language. Please, revise typographical errors throughout the manuscript.

Thank you for your comment. The authors have revised the current manuscript.

Reviewer 2 Report

Comments and Suggestions for Authors

In this paper, a methodology for optimal sensor placement is presented. This methodology incorporates a damage detection framework with simulated damage scenarios and can efficiently provide the optimal combination of sensor positions for vibration-based damage localization.

The paper is well structured and well written, and the topic is actual and interesting. The reviewer has only some minor comments to be addressed before the paper publishing:

1.       The abstract is rather long. According to MPDI prescriptions, abstracts should be at most of 200 words. The reviewer strongly suggests to reduce it also to improve its attractiveness in the reading.

2.       At the beginning of the introduction, authors discuss about the importance of VB-SHM in structures and of damage detection procedure. This part should be completed by adding references in addition to those that have already been included, which are few in number, but that are important to provide credibility to the topic. The reviewer suggests some references that can be added and that could be useful for authors to find literature about these topics:

o   Nicoletti, V., Arezzo, D., Carbonari, S., Gara, F. Detection of infill wall damage due to earthquakes from vibration data. Earthq. Eng. Struct. Dyn., 52(2), 460-481, 2023. DOI: 10.1002/eqe.3768.

o   I. Rosati, G. Fabbrocino, C. Rainieri. A discussion about the Douglas-Reid model updating method and its prospective application to continuous vibration-based SHM of a historical building. Engineering Structures, Volume 273, 2022, 115058, ISSN 0141-0296, https://doi.org/10.1016/j.engstruct.2022.115058.

o   Sneha Prasad, David Kumar, Sumit Kalra, Arpit Khandelwal. A real-time feature-based clustering approach for vibration-based SHM of large structures. Measurement, Volume 227, 2024, 114222, ISSN 0263-2241, https://doi.org/10.1016/j.measurement.2024.114222.

3.       The second part of the introduction (after the literature review) is quite long and wordy. It contains a discussion that should preferably be included in the conclusion rather than the introduction. The reviewer suggests to shorten it also to improve the readability of the paper.

4.       In Section 3, when authors address the FEM output, it is not clear how they obtain the transmittance function. Indeed, experimentally is easy to achieve, but numerically should be declared. Do the authors made some input-output numerical analyses? Please, better address this aspect.

5.       In the conclusions, authors should briefly explain how their procedure can be extended to real and more complex structures, such as buildings and bridges, also addressing possible pros and cons.

In addition, the Reviewer has other additional comments about minor typos:

1.       In the capture of Fig. 1 it is suggestable to repeat the reference in which the procedure was presented at first.

Author Response

Modifications in Response to Comments of Reviewer # 2

Answers are provided per comment

We would like to thank the reviewer for devoting his/her time to examine and study our paper and for providing constructive criticism and helpful suggestions. In preparing the revised manuscript we considered all comments, requests and suggestions, as explained next in detail.

All changes in the text are highlighted by yellow color in the revised manuscript.

  1. The abstract is rather long. According to MPDI prescriptions, abstracts should be at most of 200 words. The reviewer strongly suggests to reduce it also to improve its attractiveness in the reading.

Thank you for your comment.

The Abstract has now been revised and has a reduced length.

  1. At the beginning of the introduction, authors discuss about the importance of VB-SHM in structures and of damage detection procedure. This part should be completed by adding references in addition to those that have already been included, which are few in number, but that are important to provide credibility to the topic. The reviewer suggests some references that can be added and that could be useful for authors to find literature about these topics:
  2. Nicoletti, V., Arezzo, D., Carbonari, S., Gara, F. Detection of infill wall damage due to earthquakes from vibration data.  Eng. Struct. Dyn., 52(2), 460-481, 2023. DOI: 10.1002/eqe.3768.
  3. Rosati, G. Fabbrocino, C. Rainieri. A discussion about the Douglas-Reid model updating method and its prospective application to continuous vibration-based SHM of a historical building. Engineering Structures, Volume 273, 2022, 115058, ISSN 0141-0296, https://doi.org/10.1016/j.engstruct.2022.115058.
  4. Sneha Prasad, David Kumar, Sumit Kalra, Arpit Khandelwal. A real-time feature-based clustering approach for vibration-based SHM of large structures. Measurement, Volume 227, 2024, 114222, ISSN 0263-2241, https://doi.org/10.1016/j.measurement.2024.114222.

Thank you for your comment and for providing relative literature. The references have been added along with additional research in the revised manuscript.

  1. The second part of the introduction (after the literature review) is quite long and wordy. It contains a discussion that should preferably be included in the conclusion rather than the introduction. The reviewer suggests to shorten it also to improve the readability of the paper.

Thank you for your comment.

Some details have been excluded in the revised manuscript in the introduction section.

The intention of the authors was to provide clear understanding of the manuscript’s content. Since the methodology is incorporating a damage detection framework that has been developed and presented in the past, the authors wanted to make a clear distinction between the presented OSP methodology and the Damage Detection Framework. Also it can provide an understanding on the evolution where starting from the Framework and because of its components (TF, parametric damaged area, optimization algorithm etc.) we are able to develop this OSP methodology.

Furthermore, as it is noted in the literature review it is not common to treat the sensor locations as a combination of sensors while combining it with a damage detection method.

While the authors agree with the reviewer that some information could be excluded, the preference would be to keep it in order to provide a clear understanding of novelty and the distinction between the previous research.

If the reviewer requires it the authors can exclude the details.

  1. In Section 3, when authors address the FEM output, it is not clear how they obtain the transmittance function. Indeed, experimentally is easy to achieve, but numerically should be declared. Do the authors made some input-output numerical analyses? Please, better address this aspect.

Thank you for your comment.

The Transmittance Function is calculated between two output acceleration signals, as it is described in the Section 2.3. As such, an input signal is not required. The FE model uses frequency response analysis and a random excitation similar to the one that is applied in the experimental structure, which was referred in Section 4.1. A comment is now added in Section 4.1 in order to increase the clarity for the reader.  

  1. In the conclusions, authors should briefly explain how their procedure can be extended to real and more complex structures, such as buildings and bridges, also addressing possible pros and cons.

Thank you for your comment.

The procedure in larger or more complex structures will be identical to the one presented in the current manuscript. The only difference would be the number of different simulated damage cases that will be investigated and possibly the number of possible sensor locations (and combinations) which as mentioned before could increase the total computational cost.

A comment has been added in the conclusions to highlight this.

In addition, the Reviewer has other additional comments about minor typos:

In the capture of Fig. 1 it is suggestable to repeat the reference in which the procedure was presented at first.

Thank you for your comment. The Reference has been added to the revised manuscript.

Round 2

Reviewer 1 Report

Comments and Suggestions for Authors

the Authors revised the paper accoding to the suggestions